# Applicability of the Thrombin Generation Test to Evaluate the Hemostatic Status of Hemophilia A Patients in Daily Clinical Practice

**DOI:** 10.3390/jcm11123345

**Published:** 2022-06-10

**Authors:** Ángel Bernardo, Alberto Caro, Daniel Martínez-Carballeira, José Ramón Corte, Sonia Vázquez, Carmen Palomo-Antequera, Alfredo Andreu, Álvaro Fernández-Pardo, Julia Oto, Laura Gutiérrez, Inmaculada Soto, Pilar Medina

**Affiliations:** 1Department of Hematology, Central University Hospital of Asturias (HUCA), 33011 Oviedo, Spain; bercamez@gmail.com (A.C.); daniel_mc@hotmail.es (D.M.-C.); jrcorteb@gmail.com (J.R.C.); sonivazquez@yahoo.es (S.V.); isotor60@gmail.com (I.S.); 2Platelet Research Lab, Instituto de Investigación Sanitaria del Principado de Asturias (ISPA), 33011 Oviedo, Spain; gutierrezglaura@uniovi.es; 3Department of Internal Medicine, Central University Hospital of Asturias (HUCA), 33011 Oviedo, Spain; cpalomoantequera@gmail.com; 4Bone Metabolism, Vascular Metabolism and Chronic Inflammatory Diseases Group, Instituto de Investigación Sanitaria del Principado de Asturias (ISPA), 33011 Oviedo, Spain; 5Department of Pharmacology, University of Navarra, 31008 Pamplona, Spain; aandreures@gmail.com; 6Hemostasis, Thrombosis, Arteriosclerosis and Vascular Biology Research Group, Medical Research Institute Hospital La Fe (IIS La Fe), 46026 Valencia, Spain; alvarofernandezpardo@gmail.com (Á.F.-P.); juliaotomartinez@gmail.com (J.O.); medina_pil@gva.es (P.M.); 7Department of Medicine, University of Oviedo, 33006 Oviedo, Spain

**Keywords:** Hemophilia A, thrombin generation test, bleeding, mutation, *F8*, FVIII

## Abstract

Hemophilia A (HA) is a rare bleeding disorder caused by factor VIII (FVIII) deficiency due to various genetic mutations in the *F8* gene. The disease severity inversely correlates with the plasma levels of functional FVIII. The treatment of HA patients is based on FVIII replacement therapy, either following a prophylactic or on-demand regime, depending on the severity of the disease at diagnosis and the patient’s clinical manifestations. The hemorrhagic manifestations are widely variable amongst HA patients, who may require monitoring and treatment re-adjustment to minimize bleeding symptoms. Notably, laboratory monitoring of the FVIII activity is difficult due to a lack of sensitivity to various FVIII-related molecules, including non-factor replacement therapies. Hence, patient management is determined mainly based on clinical manifestations and patient–clinician history. Our goal was to validate the ST Genesia^®^ automated thrombin generation analyzer to quantify the relative hemostatic status in HA patients. We recruited a cohort of HA patients from the Principality of Asturias (Spain), following treatment and at a stable non-bleeding phase. The entire cohort (57 patients) had been comprehensively studied at diagnosis, including FVIII and VWF activity assays and *F8* genetic screening, and then clinically monitored until the Thrombin Generation Test (TGT) was performed. All patients were recruited prior to treatment administration, at the maximum time-window following the previous dose. Interestingly, the severe/moderate patients had a similar TGT compared to the mild patients, reflecting the non-bleeding phase of our patient cohort, regardless of the initial diagnosis (i.e., the severity of the disease), treatment regime, and FVIII activity measured at the time of the TGT. Thus, TGT parameters, especially the peak height (Peak), may reflect the actual hemostatic status of a patient more accurately compared to FVIII activity assays, which may be compromised by non-factor replacement therapies. Furthermore, our data supports the utilization of combined TGT variables, together with the severity of patient symptoms, along with the *F8* mutation type to augment the prognostic capacity of TGT. The results from this observational study suggest that TGT parameters measured with ST Genesia^®^ may represent a suitable tool to monitor the hemostatic status of patients requiring a closer follow-up and a tailored therapeutic adjustment, including other hemophilia subtypes or bleeding disorders.

## 1. Introduction

Hemophilia A (HA) is a rare X-linked coagulation disorder caused by mutations of different natures that affect the clotting factor (F) VIII gene (*F8*), leading to the protein (FVIII) deficiency. HA patients are classified based on residual plasma levels of active FVIII as: severe (<1 IU/dL), moderate (1–5 IU/dL), or mild (5–40 IU/dL) [1]. The bleeding phenotype in HA patients inversely correlates with FVIII activity, and patients with moderate and severe HA require effective treatment to overcome spontaneous joint and muscle bleeding. Conventional treatment for HA is FVIII replacement therapy, although novel therapeutic approaches are under development to reduce the treatment burden [2]. However, given similar levels of FVIII activity, the hemorrhagic manifestations are widely variable amongst HA patients, supporting the notion that FVIII activity, as a clinical measure, is not sufficient and does not reflect the hemostatic status of the patient. The complexity of the coagulation system, integrated by various pro-and anticoagulant and pro-and antifibrinolytic pathways interacting with each other, make FVIII activity alone inadequate to use as a prognostic variable [3,4]. Alterations in some of these pathways in patients with HA may cause different hemorrhagic phenotypes. Additionally, some mutations may explain the non-bleeding or mild phenotype of some severe HA patients due to an acquired enhanced affinity of FVIII for FX, as previously described for the variant p.Arg1781His [5].

Thrombin is the central factor in the coagulation process, and proper hemostasis requires a precise balance between clot formation and dissolution [6]. An assay determining the ability of plasma to generate thrombin could be applied to infer the hemostatic status of HA patients [7]. The thrombin generation test (TGT) is a global coagulation assay assessing the capacity of a given plasma sample to generate thrombin, in a physiologically relevant milieu [8,9,10,11,12,13,14,15]. TGT has demonstrated usefulness to stratify bleeding risk in patients with severe HA with greater precision compared to FVIII activity [16,17,18,19,20]. Current FVIII activity test methodologies are not sensitive enough to detect the various FVIII-related molecules used for replacement therapy, including nonfactor agents [21]. The basis of TGT consists in the activation of plasma coagulation by small/limiting amounts of tissue factor and phospholipids, and the continuous monitoring of thrombin formation using a specific fluorogenic substrate. The most widely utilized parameters derived from the thrombin generation curve are the endogenous thrombin potential (ETP) and the peak height (Peak). ETP, expressed in nmol*min, is the net amount of thrombin generated in plasma on the basis of the relative strength of the pro- and anticoagulant drivers and the time of the reaction. Peak, expressed in nmol, is defined as the maximum amount of thrombin formed during the reaction. Other parameters include the Lag Time, which is the time corresponding to 1/6th of the Peak height, reflecting the approximate clotting time; the Time to Peak, which is the time the reaction taking to reach the Peak; and the Start Tail, which is the time the reaction takes to reach the end point (Figure 1) [22]. The Velocity Index, which indicates the velocity of active thrombin formation (nmol/min), is another informative parameter, however, it is the parameter with more inter-individual variation, even in normal subjects [23]. The calibrated automated thrombogram (CAT) is one commercially available assay to perform TGT. However, CAT and other similar assays have limitations related to standardization and quality control. Furthermore, they are not fully automated, making the simultaneous analysis of individual plasmas challenging in routine clinical settings [24,25].

More recently, the development of the ST Genesia^®^ platform, a fully automated device, has allowed improved reproducibility compared to semi-automated methods, due to the inclusion of a reference plasma for result normalization, improved temperature control, and the reagents intended for use on the platform [26,27,28].

In the present study, we performed TGT measurements in HA patients using the ST Genesia^®^, with the aim of assessing its diagnostic and prognostic performance. We correlated the outcomes obtained with the clinical manifestations of the disease and FVIII activity levels, annual bleeding rate (ABR), identified mutations, and treatment regime. 

## 2. Materials and Methods

### 2.1. Patient Cohort and Healthy Controls of the Study

The entire cohort of 57 HA outpatients in a stabilized, non-bleeding phase while under treatment were enrolled in the Central University Hospital of Asturias (HUCA, Oviedo, Spain) between 2018 and 2019. Objective diagnosis of HA was made considering FVIII activity as measured with one stage FVIII clotting assay (FVIII:C, Hemosil) and chromogenic FVIII clotting assay (FVIII:Chr, Electrachrome). All had FVIII (FVIII:C) activity below 40% at diagnosis. One patient with FVIII activity above 40% carries a confirmed *F8* variant.

To rule out the presence of von Willebrand Disease (VWD), antigenic VWF (VWF:Ag) and ristocetin cofactor activity (VWF:RCo) were performed. For each patient, the annual bleeding rate (ABR) (bleeds/year) was calculated as the number of bleeding events (spontaneous and traumatic) during 2 consecutive years divided by 2. ABR was calculated for 2017–2018 (preTGT), and 2018–2019 (postTGT).

The exclusion criteria included: presence of other coagulopathies, such as VWD; deficiency of other coagulation factors; acquired hemophilia; and development of inhibitors. Three patients developed inhibitors and were excluded from the study.

Age, gender, and comorbidity-matched 25 unrelated Caucasian healthy subjects with no history of HA or HB from the same geographical area were included in the study as control group. All assays and measurements were performed in the same clinical lab where recruitment was undertaken (HUCA, Oviedo, Spain).

All subjects participated after giving written or verbal informed consent according to protocols approved by the Medical Ethics Committee of our Institution and the Declaration of Helsinki, as amended in Edinburgh in 2000.

### 2.2. Sample Preparation

Blood samples were collected in citrate-anticoagulated Vacutainer tubes (BD Diagnostics). Plasma was obtained by centrifugation at 1.811× *g* for 30 min at 4 °C and stored in aliquots at −80 °C until further use. For genetic studies, blood was collected in EDTA-anticoagulated 5 mL tubes. All patients following prophylactic treatment with factor concentrate were recruited prior the administration of the following dose (i.e., 48–72 h after the previous dose). The rest of patients had not received treatment for the previous 7 days.

### 2.3. Thrombin Generation Test

TGT parameters were calculated using the ST Genesia^®^ (Stago, Asnieres-sur-Seine, France), using STG^®^-BleedScreen (STG-BLS). The manufacturer´s protocol was followed without modification. Procoagulant phospholipids and low picomolar levels of human tissue factor (TF) were added according to the manufacturer´s protocol. The assay contained two quality controls for low and normal TG activity, respectively, along with a reference plasma for normalization of parameters. Normalized data with ETP and Peak represented as percentage (%), and Time to Peak, Lag Time, and Start Tail, as ratio of the respective values of control plasma were used for statistical analyses.

### 2.4. Identification of F8 Mutations

Genomic DNA was isolated by using the QIAamp DNA Blood Mini Kit (QIAGEN, Hilden, Germany), following manufacturer´s instructions. The DNA concentration and purity was measured in a NanoDrop ND-1000 spectrophotometer (Thermo Fisher Scientific, Waltham, MA, USA). Sequencing of *F8* exons and flanking regions was conducted using TruSight™ One (Illumina, San Diego, CA, USA) and Nextera Rapid Capture™ technology in a NextSeq™ sequencer (Illumina). *F8* variants were identified by alignment with the human reference genome GRCh37 (hg19) of the Genome Reference Consortium after filtering, using specific quality criteria.

Variant annotation was performed using Alamut Visual™ (Interactive Biosoftware, Rouen, France), Variant Interpreter™ (Illumina) and Ingenuity Variant Analysis™ (QIAGEN), using as reference the population databases dbSNP (www.ncbi.nlm.nih.gov/snp/), 1000 Genomes (www.internationalgenome.org/1000-genomes-browsers/), ExAC (exac.broadinstitute.org), gnomAD (gnomad.broadinstitute.org), Human Gene Mutation Database (HGMD (QIAGEN); Cardiff, UK) (www.hgmd.cf.ac.uk/ac/index.php), ClinVar (www.ncbi.nlm.nih.gov/clinvar/), and LOVD (www.lovd.nl); all accessed on 1 January 2021.

Pathogenicity of the identified variants was analyzed in silico using Mutation Taster (http://www.mutationtaster.org/), SIFT (sift.bii.a-star.edu.sg) and Poly-Phen-2 (genetics.bwh.harvard.edu/pph2/); all accessed on 1 January 2021.

The identified variants were named according to the Human Genome Variation Society (HGSV) (www.hgvs.org/mutnomen/, accessed on 1 January 2021) and classified based on the recommendations of the American College of Medical Genetics and Genomics (ACMG) [29].

When no variant was identified, the DNA samples were analyzed for detection of large rearrangements and large deletions/duplications (Copy Number Variant; CNV) using the SALSA MLPA Probemix P178 *F8* (MRC Holland), following manufacturer´s instructions. Additionally, insertions/deletions of ≥10 nucleotides, mutations in repetitive regions and variants in pseudogenes with highly homologous sequences were screened by direct sequencing using specific primers designed to overcome these difficulties and high fidelity Taq polymerase (primer set sequences are available upon request).

### 2.5. Statistical Analysis

The distribution of the data was assessed with the Shapiro–Wilk test. Continuous variables were expressed as median and 1st–3rd quartile (25–75%) and categorical variables were presented as count and percentage. *t* test of paired samples, non-parametric tests of independent samples (Mann–Whitney U test or the related Kruskal–Wallis, with Bonferroni correction for multiple comparisons) and Spearman correlation test were performed for statistical analysis with SPSS package (v.27, IBM SPSS Statistics, Armonk NY, USA). Graphs were prepared using BoxPlotR (http://shiny.chemgrid.org/boxplotr/, accessed on 1 January 2021) [30] and Adobe Illustrator CS4. *p*-values < 0.05 were considered statistically significant.

## 3. Results

### 3.1. Diagnosis and Epidemiological Characteristics of the HA Cohort of Study

The complete cohort included 54 HA patients, of which 18 presented with severe HA (33.3%), 7 patients presented with moderate HA (13.0%), and 29 patients presented with mild HA (53.7%). Diagnoses were done based on results from FVIII:Chr and FVIII:C activity assays (Figure 2A, Table 1 and Appendix A). Given the low number of patients with moderate HA, we grouped together severe and moderate HA patients for the majority of comparisons, after confirming no significant differences between severe and moderate patients for any of the variables analyzed (data not shown). Compared to mild HA patients, patients with severe/moderate HA had a significantly lower FVIII activity as measured both by FVIII:C and FVIII:Chr assays at diagnosis and an increased annual bleeding rate (ABR preTGT) (Table 1). No significant differences were observed in VWF:Ag or VWF:RCo levels amongst patients with severe/moderate HA compared to mild HA patients at diagnosis (Table 1). Treatment regimens are depicted in Table 1 for the HA patient groups (see also Appendix A).

Of the 54 HA patients, 25 experienced at least one bleeding episode between 2017 and 2018 (ABR preTGT), of which 7 patients experienced 3 or more bleeding episodes, and 3 patients experienced a thrombotic event during the aforementioned period. In the time period of the study, 11 HA patients underwent surgery: 1 severe HA patient experienced one bleeding episode; 1 moderate HA patient experienced 6 bleeding episodes; and of the 9 mild HA patients, 3 experienced 1 bleeding episode. Regarding viral infections, 14 HA patients (10 severe and 4 mild) had hepatitis B, 23 HA patients (10 severe, 2 moderate, and 11 mild) had hepatitis C, and 8 HA patients (7 severe and 1 mild) had HIV. The main comorbidities registered were: smoking (10 patients), dyslipidemia (11), cirrhosis (7), arterial hypertension (7), alcohol (5), diabetes mellitus type 2 (2), and obesity (2) (Appendix A).

As shown in Figure 2B, and as anticipated, severity significantly correlated with FVIII activity as measured by FVIII:Chr and FVIII:C assays and with the ABR calculated in the period 2017–2018 (ABR preTGT). Supporting previous reports [31,32], neither VWF:Ag nor VWF:RCo levels correlated with HA severity (Figure 2B).

The genetic study performed for the entire cohort identified 27 mutations of different natures in the *F8* gene of 45 patients; most of them were previously identified and associated with HA severity (Figure 2C). Of these variants, five were not previously described [33], but have since been registered at The European Association for Haemophilia and Allied Disorders (EAHAD) Coagulation Factor Variant Databases (Factor VIII) [34]. Neither mutations nor large insertions or deletions were identified in the remaining 10 patients (18%), by sequencing or by MPLA, respectively.

### 3.2. Clinical and Epidemiological Characteristics of the HA Cohort of Study, at a Non-Bleeding Phase, and at the Time of TGT

At the time of TGT measurement, all HA patients were in a stable, non-bleeding phase, with a settled personalized treatment regime (Table 1, Table 2 and Appendix A). As the control, we recruited 25 male age- and comorbidity-matched healthy donors (Table 1, Table 2 and Appendix A). Of note, there was a significant difference regarding age, affecting only the severe/moderate HA group when compared to the control group (Table 2). In patients under treatment, measurements were performed during the maximum time window from the previous dose administration, depending on treatment regime, as explained in the methods section. We measured FVIII activity using the appropriate assay according to the treatment regime, following the WFH Guidelines for the Management of Hemophilia [35]. FVIII activity in the controls was measured using the FVIII:C assay and was within the normal range (Table 2). Regarding the patients, at the time of recruitment, the measured FVIII activity corresponded to the HA severity, but was significantly increased compared to activity assessed at diagnosis (Figure 3 and Table 2). We should consider that patients were recruited at the “valley” phase, prior treatment administration, and that laboratory tests are not always suitable for FVIII activity studies as they may not detect all FVIII-related molecules. 

Of the 54 HA patients, 20 experienced a bleeding episode between 2018 and 2019 (ABR postTGT, with only two patients who experienced three or more bleeding episodes), and only one patient with mild HA experienced a thrombotic event (myocardial infarction) during this period. The ABR postTGT was significantly reduced in the mild HA group, compared to the ABR preTGT (Table 2).

### 3.3. TGT Parameters of the HA Cohort of Study, at a Non-Bleeding Phase

The normalized results of the TGT parameters are shown in Table 3 (see also Appendix A). All TGT variables in the HA patient groups (except for Lag Time) differed significantly versus the controls, but not amongst the HA patient groups (Figure 3 and Table 3).

The parameters most drastically decreased were Peak (69–72%) and ETP (47–49%) (Figure 3 and Table 3). These results indicate that stabilized, non-bleeding severe/moderate HA patients have a similar capacity to generate thrombin versus mild HA patients, despite having a lower plasma FVIII activity on the same day as measured by FVIII:C and FVIII:Chr assays. Thus, TGT, rather than FVIII:C or FVIII:Chr assays, may represent a more reliable tool to monitor the hemostatic competence of HA patients.

Based on our findings, while FVIII activity highly correlates with disease severity, it does not appear to be an accurate marker of the hemostatic status of HA outpatients in a stabilized, non-bleeding phase of the disease, since the ability of severe/moderate HA patients to generate thrombin is similar to that of mild HA patients.

To get a deeper insight into the interpretation of TGT variables, we next performed correlation analysis on controls and HA patients separately (Figure 4). Interestingly, while FVIII activity significantly correlates with ETP in the controls, the correlation was lost in HA patients, supporting the notion that TGT parameters, rather than FVIII activity, accurately indicate the patients’ hemostatic status. Additionally, other correlations acquired significance in HA patients as compared to the controls, linking FVIII activity to Peak, Lag Time, and Start Tail. While severity, ABR, and FVIII activity correlated significantly, neither ABR nor Severity correlated with TGT parameters (except for ABR postTGT with Start Tail). Importantly, the changes in the correlation of TGT parameters themselves demonstrate the potential sensitivity of the platform. In the controls, ETP correlates directly with Start Tail, but in HA patients, the ETP inversely correlates, meaning the reaction is exhausted earlier in HA patients. A similar change in direction is observed with Lag Time and Peak, shifting from an inverse (controls) to a direct (HA) correlation, meaning the reaction tends to be slower in HA patients.

### 3.4. TGT Parameters of the HA Cohort of Study, at a Non-Bleeding Phase, Based on ABR (PreTGT)

These observations prompted us to study TGT parameters on HA patients, stratified based on their registered bleeding episodes (Table 4). As shown in Table 4, there were no significant differences observed in TGT parameters when comparing patients with none or at least one bleeding episode (ABR preTGT), while the FVIII activity differed significantly. As mentioned before, a tendency to reduction of bleeding episodes was observed when comparing the ABR preTGT with the ABR postTGT, which was determined statistically significant in the group of mild HA patients (Table 1, Table 2 and Table 3 and S1). These results suggest the TGT parameters may have prognostic value, as the ABR postTGT showed a tendency of reduction in the number of patients experiencing a bleeding episode, but also on the number of bleeding episodes experienced (see Appendix A).

### 3.5. TGT Parameters of the HA Cohort of Study, at a Non-Bleeding Phase, Based on ABR (PreTGT) and Mutation

The mutation type has been associated with disease severity. To analyze the degree of correlation between the genotype and phenotype, we stratified HA patients based on the ABR score and identified mutations in three groups, as specified in Table 5A. Group 1 comprised patients with an ABR ≥ 1 and an identified mutation. Group 2 comprised patients with an ABR = 0 and an identified mutation. Group 3 comprised patients with no identified mutation.

As seen in Table 5B, patients in Group 1 showed, besides a higher ABR, a higher proportion of severe patients, and had a lower FVIII activity compared to those without identified mutations (Group 3). The Peak and Lag Time were also significantly reduced in Group 1 compared to Group 3. TGT parameters did not reach significance amongst Group 1 or Group 3 and Group 2 patients. Therefore, a special caution should be taken in HA patients bearing the mutations included in Group 1 (Table 5A), with increased ABR and reduced TGT parameters, such as Peak and Lag Time, compared to patients with milder manifestations of the disease. 

### 3.6. TGT Parameters of the HA Cohort of Study, at a Non-Bleeding Phase, Based on Treatment

We also analyzed the different parameters studied according to the type of treatment received (Elocta^®^, Nuwiq^®^, or Kovaltry^®^). Importantly, treatment was based on FVIII levels at diagnosis and particular patient aspects (TGT never influenced the choice of treatment). At diagnosis, patients treated with Elocta^®^ had the lowest FVIII activity and were also the patients with the highest proportion of severity. Patients treated with Kovaltry^®^ had an intermediate FVIII activity, and those treated with Nuwiq^®^ had the highest FVIII activity. Remarkably, patients treated with Nuwiq^®^ had the highest ETP and Peak values, reaching close-to-normal levels, and a lower Start Tail than those who received the other two FVIII concentrates. However, differences did not reach significance, probably due to the small number of patients in the groups of patients treated with Nuwiq^®^ and Kovaltry^®^ (Table 6). However, we should consider that the comparison amongst treatment groups may be biased by the inherent half-life differences between concentrates.

## 4. Discussion

HA patients suffer spontaneous bleeding, with intracranial hemorrhages being the leading cause of bleeding-related death in hemophilia patients [36]. The major cause of disability from bleeding is chronic joint disease [37], although additional complications are also frequently identified [38]. Current treatment with FVIII concentrates has improved life expectancy, ameliorating chronic joint disease in HA patients, and improving their quality of life. However, given similar levels of FVIII activity, the hemorrhagic manifestations are widely variable amongst HA patients, supporting the notion that FVIII activity, as a clinical measure, is not sufficient and does not reflect the hemostatic status of the patient, making it inadequate to use as a prognostic variable [3,4]. This apparent discrepancy may arise as a consequence of alterations in the finely tuned hemostatic balance between pro- and anticoagulant proteins involved in the coagulation cascade, along with inhibitors or fibrinolytic components. As a result, FVIII:C or FVIII:Chr levels do not seem to be sufficiently informative to identify those HA patients at a higher bleeding risk, especially for those patients with mild or moderate HA. As employed in other clinical situations [39,40,41,42,43], the TGT may represent a valuable tool to evaluate the hemostatic status of HA patients [16,17,18,19,20], in a more physiologically relevant manner, provided it monitors the capacity of the patient´s plasma to generate thrombin. However, previous studies have utilized TGT platforms despite the acknowledged limitations around the lack of standardization and automation, narrowing the application of findings using those platforms to highly specialized, low N settings.

In our study, we explored the utility of the fully automated ST Genesia^®^ to evaluate the hemostatic status of the entire cohort of 54 HA outpatients following (on demand and prophylactic) treatment, in a stabilized, non-bleeding period in the Principality of Asturias (Spain).

As expected, all HA patients had a significantly lower FVIII activity compared to the healthy controls, and patients with severe/moderate HA had significantly lower FVIII activity at diagnosis and a higher ABR versus mild HA patients. On the day of TGT, patients with severe/moderate HA still showed a significantly lower FVIII activity compared to mild HA patients, although FVIII activity was significantly higher compared to the activity observed at the time of diagnosis. Of note, patients were recruited for TGT at the longest time-window since their last treatment dose, and laboratory assays do not always reflect physiologically relevant, therapeutic FVIII activity status. Surprisingly, despite slight decreases in ETP observed in severe/moderate HA patients, no significant differences were observed in TGT parameters when comparing mild HA to severe/moderate HA pooled data. Predictably, all HA patients had a significantly lower ETP and Peak, and a significantly higher Time to Peak and Start Tail compared to the healthy controls but were similar amongst themselves.

Our results reveal that severe/moderate HA patients in a stabilized, non-bleeding phase of the disease have a similar capacity to generate thrombin compared to mild HA patients, regardless of their lower FVIII activity, which may explain the absence of spontaneous bleedings in severe/moderate patients. Consequently, TGT may represent a more suitable tool to ascertain the hemostatic status of HA patients versus FVIII levels. Accordingly, the ST Genesia^®^ may facilitate the identification of HA patients at a high bleeding risk requiring a closer follow-up and the therapeutic adjustment of dose and/or frequency of prophylactic treatment, especially in high-risk situations, such as patients desiring higher levels of physical activity and participation in competitive sports. Importantly, since it is a fully automated system, it enables the comparison of results among different laboratories, provided that pre-analytical and analytical conditions/protocols are followed [44]. Remarkably, the parameters that most drastically decreased in our HA patients were Peak and ETP (Table 3). Thus, these parameters might be the most relevant to ascertain the bleeding risk in HA patients. In addition, we evaluated the suitability of the clinical variables recorded and TGT as predictors of clinical phenotype in HA patients. Those HA patients with at least one previous bleeding episode had a lower FVIII activity at diagnosis. Previous studies have analyzed the role of TGT in the clinical evaluation of HA patients [10,16,17,18,19,20,45,46,47], demonstrating that TGT parameters tend to be decreased and delayed in HA patients compared to healthy individuals, and our results corroborate this notion. However, to the best of our knowledge, this is the largest cohort of HA outpatients in which the ST Genesia^®^ has been evaluated, a cohort that includes patients with different treatment regimens and FVIII-related molecules, including non-factor replacement therapies. While the data presented here are preliminary in nature, they support the notion that TGT is a useful prognostic tool to assess the risk of future bleedings in HA patients, and future efforts should determine the TGT prognostic capacity on a more complete follow-up and pharmacokinetic studies.

The underlying mutation in the X-linked *F8* is the most important determinant of residual FVIII activity in HA patients [48,49]. Gene inversions account for approximately 45% of the *F8* pathogenic variants in severe HA, the most common locating to intron 1 and 22 [50,51,52]. Up to ~5% of HA patients carry deletions and duplications in *F8* [53,54,55]. In our study, we identified 27 different *F8* mutations in 82% of HA patients, five of them novel, while no mutation or large insertions or deletions were identified in the remaining 18% of patients (all mild HA patients). When stratifying HA patients based on the identified mutation and ABR score (Table 5), we observed that Group 1 patients, besides a higher ABR, had a higher severity, a lower FVIII activity at diagnosis, and a lower thrombin potential (Peak and Lag Time) than those patients without identified mutations (Group 3). These results support the notion, as has been largely acknowledged, that the mutation type, in combination with clinical and laboratory parameters, may aid in the prognosis of bleeding risk in HA patients [20,56,57].

Provided that FVIII is protected against degradation in the circulation by its carrier molecule VWF [58,59], several reports have suggested that VWF levels may influence the HA phenotype through the association with FVIII activity [60,61,62]. We found no differences in VWF:Ag or VWF:RCo levels among HA patients. Furthermore, no correlation was identified between VWF:Ag or VWF:RCo levels and HA severity. In line with our results, Rejto et al. [32] and Loomans et al. [31] found no association between FVIII and VWF levels in mild HA patients in a stabilized, non-bleeding phase of the disease. However, no correlation with severity could be performed since neither severe nor moderate HA were studied.

Last, we observed no significant differences on TGT performance when stratifying patients based on treatment, although the number of patients included per group is limited in number. Remarkably, TGT performance reached almost normal values on those patients treated with Nuwiq^®^. This is an aspect that should be further studied recruiting larger cohorts.

A limitation of our study is the current absence of patients with active hemorrhagic manifestations or the analysis of thrombophilia markers in our patient cohort, which may influence the TGT results. However, previous reports have demonstrated thrombophilia does not protect against severe bleeding [63]. Nonetheless, although several efforts have been made [26], additional multi-center studies are needed to determine the normal TGT values in a variety of clinical conditions. Of note, we carried all sample centrifugations at 4^o^C; however, it is acknowledged that they should be done at room temperature [44]. We are certain that this has not affected the results, as controls and patient samples were handled in the same way, but we have updated our protocol since. Additionally, we should consider that the ability of a TGT to discern hemophilia is dependent on the concentration of TF used. As one of the provided reagents of the ST Genesia^®^ platform, the concentration of TF remains a manufacturing secret; however, the amount of TF in the provided reagent has been optimized in order to minimize the potential procoagulant interference from contact phase activation during blood draw.

## 5. Conclusions

Efforts are needed to understand the mechanisms preventing severe HA patients from bleeding despite having a persistently low FVIII activity. An imbalance in the finely tuned coagulation–anticoagulation–fibrinolysis equilibrium may contribute to this unexpected phenotype, and the evaluation of the hemostatic status of each patient may aid in identifying HA patients at a high-and low-bleeding risk more precisely compared to active FVIII levels. We provide here relevant evidence that the fully automated ST Genesia^®^ platform may be a valuable tool to ascertain the hemostatic status of HA patients, which may be important to predict bleeding risk regardless of FVIII activity, by studying a complete cohort of HA outpatients from Asturias (Spain) in a stabilized, non-bleeding phase of the disease. Interestingly, our data substantiate the notion that the *F8* mutation type may be a prognostic indicator of bleeding rates and the hemostatic condition [20,56,57]. The identification of high- and low-risk patients may anticipate acute bleeding incidents, prompting a closer follow-up, thus improving patient care and alleviating suffering. Our results suggest TGT parameters measured using the ST Genesia^®^ platform may represent a suitable tool to monitor the hemostatic status of patients requiring a closer follow-up and a tailored therapeutic adjustment, overcoming the limitations of FVIII activity laboratory tests in clinical practice, and its application may be extended to other types of hemophilia or bleeding disorders.

## Figures and Tables

**Figure 1 jcm-11-03345-f001:**
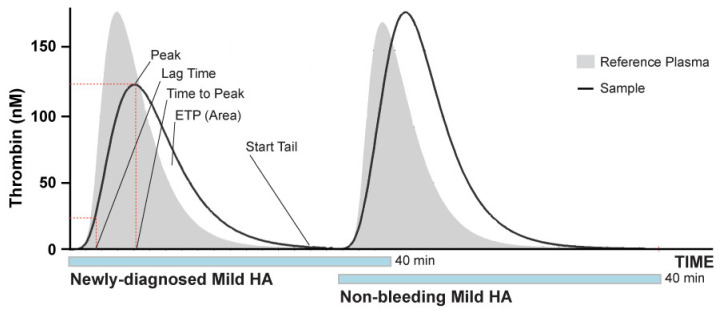
Schematic representation of TGT curves generated by the ST Genesia^®^. Representative graphs of TGTs from samples obtained from mild HA patients, a newly diagnosed patient (left), and a previously diagnosed patient without current bleeding symptoms (right). The TGT variables are indicated.

**Figure 2 jcm-11-03345-f002:**
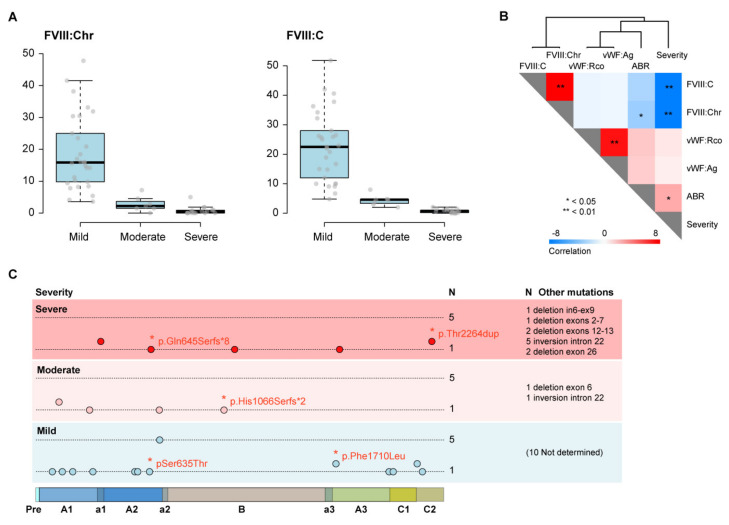
Characterization of the Hemophilia A cohort. (**A**) Box plots depicting the FVIII activity levels at diagnosis, as measured by FVIII:Chr (left) and FVIII:C (right) assays in mild, moderate, and severe HA patients. (**B**) Correlation analysis of FVIII activity levels (measured using FVIII:Chr and FVIII:C assays), VWF:Ag (antigenic Von Willebrand Factor levels), VWF:RCo (Von Willebrand Factor ristocetin cofactor activity), severity, and annual bleeding rate (ABR preTGT) in HA patients. Grades of blue represent inverse correlation and grades of red represent direct correlation. The level of significance is indicated with asterisks (Spearman correlation, bilateral significance). (**C**) Schematic representation of FVIII mutations and variants across the FVIII precursor sequence, as identified in the entire HA cohort, separated by severity. The new variants are indicated in red. No mutations were identified in 10 mild HA patients. The number of patients bearing a given mutation is represented (scale *n* = 1–5). In, intron; ex, exon.

**Figure 3 jcm-11-03345-f003:**
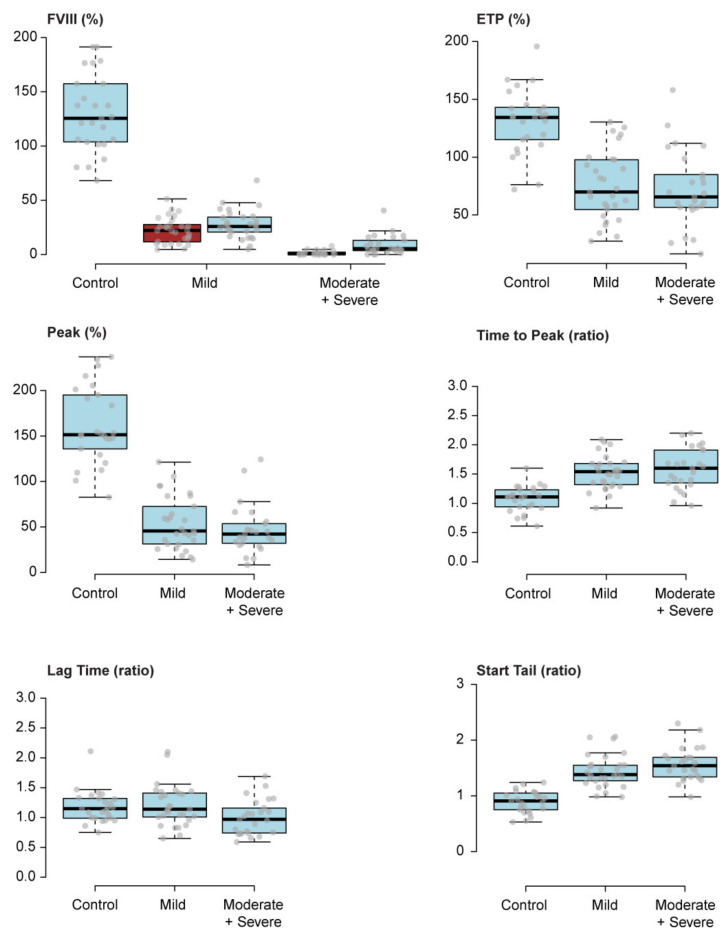
Box plots depicting TGT parameters and FVIII activity at the time of TGT. FVIII activity measured at the time of TGT and at the time of diagnosis (matched assay) in brown (upper left graph). In the other graphs, Endogenous Thrombin Potential (ETP); Peak; Time to Peak; Lag Time and Start Tail are depicted.

**Figure 4 jcm-11-03345-f004:**
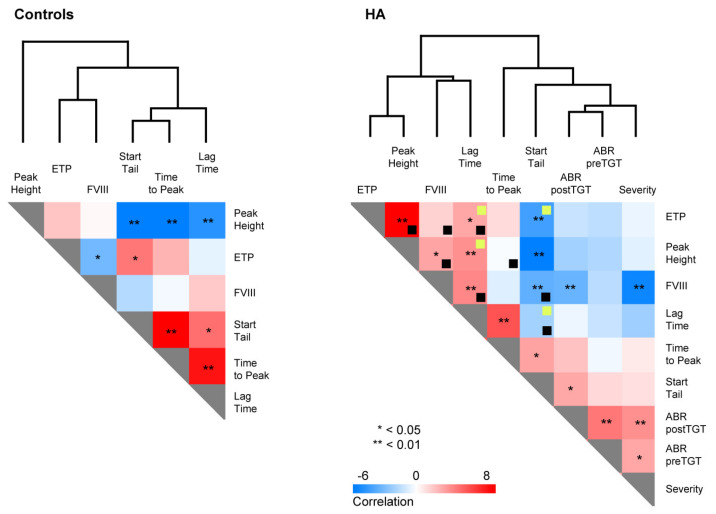
Correlation of TGT and clinical parameters. Correlation analysis of FVIII activity levels and TGT parameters in control subjects (left) and correlation analysis of FVIII activity, annual bleeding rates (ABR, preTGT, and postTGT), severity, and TGT parameters in HA patients (right). Grades of blue represent inverse correlation and grades of red represent direct correlation. The level of significance is indicated with asterisks (Spearman correlation, bilateral significance). Black squares indicate a change of significance (significant–not significant) as observed in control correlations, and the yellow squares indicate a change in the sign of the correlation (inverse–direct).

**Table 1 jcm-11-03345-t001:** Clinical characterization of the HA cohort of study at diagnosis.

	Mild HA *n* = 29 (A)	Severe + Moderate HA *n* = 25 (B)	*p*-Value * A vs. B
FVIII:Chr (%)	15.9 (9.6–27.7)	0.7 (0.0–1.8)	**<0.001**
FVIII:C (%)	22.5 (11.0–30.1)	1.0 (0.3–2.8)	**<0.001**
**FVIII:Chr vs. FVIII:C** **(paired)**	** *p* ** **-value A** **<0.001**	** *p* ** **-value B** **NS**	
VWF:Ag (%)	120.8 (89.1–140.0)	119.5 (88.9–183.4)	NS
VWF:RCo (%)	94.2 (76.0–117.6)	97.6 (83.3–114.0)	NS
ABR (preTGT)	0.0 (0.0–1.0)	1.0 (0.0–2.0)	**<0.01**
Treatment regimes			
Prophylactic	1 (3.4)	17 (68.0) 8 (32.0)	
On demand	28 (96.6)		
Elocta^®^			
Prophylactic	1 (3.4)	14 (56.0) 5 (20.0)	
On demand	0 (0.0)		
Nuwiq^®^			
Prophylactic	0 (0.0)	3 (12.0) 0 (0.0)	
On demand	3 (10.3)		
Kovaltry^®^			
Prophylactic	0 (0.0)	0 (0.0) 2 (8.0)	
On demand	2 (6.9)		
Fandhi^®^			
Prophylactic	0 (0.0)	0 (0.0) 0 (0.0)	
On demand	1 (3.4)		
Advate^®^			
Prophylactic	0 (0.0)	0 (0.0) 1 (4.0)	
On demand	0 (0.0)		
DDAVP			
Prophylactic	0 (0.0)	0 (0.0) 0 (0.0)	
On demand	17 (58.6)		
DDAVP/Advate^®^			
Prophylactic	0 (0.0)	0 (0.0) 0 (0.0)	
On demand	1 (3.4)		
DDAVP/Fandhi^®^			
Prophylactic	0 (0.0)	0 (0.0) 0 (0.0)	
On demand	1 (3.4)		
DDAVP/Nuwiq^®^			
Prophylactic	0 (0.0)	0 (0.0) 0 (0.0)	
On demand	3 (10.3)		

Continuous variables are presented as median and 1st–3rd quartile. Categorical variables are presented as count and percentage. * Mann–Whitney U tests of independent samples or *t*-test of paired samples were performed; significant results are in bold. HA, hemophilia A; FVIII:C, plasma FVIII concentration; FVIII:Chr, chromogenic FVIII levels; TGT, thrombin generation test; ABR: annual bleeding rate; VWF:Ag, antigenic Von Willebrand Factor levels; VWF:RCo, Von Willebrand Factor ristocetin cofactor activity; NS, not significant.

**Table 2 jcm-11-03345-t002:** Clinical characterization of the HA cohort and healthy controls at the time of TGT.

	Controls *n* = 25 (A)	Mild HA *n* = 29 (B)	Severe + Moderate HA *n* = 25 (C)	*p*-Value * A vs. B	*p*-Value * A vs. C	*p*-Value * B vs. C
Age, years	61.0 (41.0–70.0)	40.0 (21.5–61.5)	34.0 (26.5–46.5)	NS	**<0.01**	NS
FVIII (Chr or C; %)						
^$^ At TGT	125.6 (102.7–157.5)	25.9 (18.8–34.9)	5.5 (3.5–14.6)	**<0.001**	**<0.001**	**= 0.001**
TGT Chr		14.6 (*n* = 1)	6.7 (3.0–16.1) (*n* = 18)	ND	ND	ND
TGT C	125.6 (102.7–157.5)	26.8 (20.9–35.0) (*n*= 28)	4.6 (4.3–6.1) (*n*= 7)	**<0.001**	**<0.001**	NS
^&^ At diagnosis (assay matched)	ND	22.5 (11.0–30.1)	0.9 (0.0–2.6)	ND	ND	**<0.001**
**FVIII TGT ^$^ vs. diagnosis ^&^** **(paired)**	ND	** *p* ** **-value B** **<0.05**	** *p* ** **-value C** **<0.001**			
ABR preTGT, No.	ND	0.0 (0.0–1.0)	1.0 (0.0–2.0)	ND	ND	**<0.01**
ABR postTGT, No	ND	0.0 (0.0–0.0)	1.0 (0.0–1.8)	ND	ND	**<0.001**
**ABR pre vs. post TGT** **(paired)**	ND	** *p* ** **-value B** **<0.05**	***p*****-value C** NS			

Continuous variables are presented as median and 1st–3rd quartile. Categorical variables are presented as count and percentage. FVIII activity measured at the day of TGT is depicted, and the FVIII activity levels at diagnosis (matching the assay used) is also shown. * Kruskal–Wallis tests of independent samples (Bonferroni correction for multiple comparisons was done for the variables with more than 2 groups being compared) or *t*-tests of paired samples were performed, significant results are in bold. HA, hemophilia A; FVIII:C, plasma FVIII concentration; FVIII:Chr, chromogenic FVIII levels; TGT, thrombin generation test; ABR: annual bleeding rate; ND, not determined; NS, not significant; $, measured at the time of TGT; &, measured at the time of diagnosis.

**Table 3 jcm-11-03345-t003:** Thrombin generation parameters in the HA patient cohort and healthy controls.

	Controls *n* = 25 (A)	Mild HA *n* = 29 (B)	Severe + Moderate HA *n* = 25 (C)	*p*-Value * A vs. B	*p*-Value * A vs. C	*p*-Value * B vs. C
ETP (%)	134.3 (113.0–144.1)	70.0 (51.9–97.9)	65.5 (56.4–91.7)	**<0.001**	**<0.001**	NS
Peak (%)	151.4 (132.6–198.1)	45.6 (31.2–78.3)	42.2 (31.0–54.7)	**<0.001**	**<0.001**	NS
Time to peak(ratio)	1.1 (0.9–1.2)	1.5 (1.3–1.7)	1.6 (1.3–1.9)	**<0.001**	**<0.001**	NS
Lag time (ratio)	1.2 (1.0–1.3)	1.1 (1.0–1.4)	1.0 (0.7–1.2)	NS	NS	NS
Start tail (ratio)	0.9 (0.7–1.1)	1.4 (1.3–1.6)	1.5 (1.3–1.7)	**<0.001**	**<0.001**	NS

Continuous variables are presented as median and 1st–3rd quartile. * Kruskal–Wallis test of independent samples (Bonferroni correction for multiple comparisons) was performed, significant results are in bold. HA, hemophilia A; ETP, endogenous thrombin potential. NS, not significant.

**Table 4 jcm-11-03345-t004:** Clinical characteristics of the HA patients according to the ABR preTGT.

Characteristics	ABR = 0 (*n* = 29)	ABR > 0 (*n* = 25)	Statistical Significance (*p*) *
Severity **			
**1**	21 (72.4)	8 (32)	
**2**	1 (3.4)	6 (24)	
**3**	7 (24.1)	11 (44)	
Age, years	35 (20.5–48.0)	37.0 (29.5–53.0)	NS
FVIII TGT	21.8 (10.6–28.1)	10.3 (4.3–27.7)	NS
Paired FVIII diagnosis	16.7 (3.3–25.8)	3.2 (0.2–9.5)	**<0.05**
VWF:Ag (%)	111.0 (89.6–136.3)	121.8 (88.6–186.0)	NS
VWF:RCo (%)	94.2 (71.2–107.0)	99.0 (84.3–134.6)	NS
ETP (%)	72.8 (57.3–99.0)	65.5 (49.2–89.7)	NS
Peak (%)	47.1 (35.6–69.3)	38.3 (29.3–60.1)	NS
Time to peak (ratio)	1.6 (1.3–2.0)	1.5 (1.3–1.7)	NS
Lag time (ratio)	1.1 (0.9–1.4)	1.0 (0.7–1.3)	NS
Start tail (ratio)	1.4 (1.3–1.7)	1.5 (1.3–1.7)	NS

Continuous variables are presented as median and 1st–3rd quartile. FVIII activity as measured at the time of TGT, and at diagnosis (assay matched). VWF:Ag and VWF:RCo levels were measured at diagnosis. * Mann–Whitney non-parametric U Test (Bonferroni correction for multiple comparisons), significant results are in bold. ** Severity: 1 Mild; 2 Moderate; 3 Severe. HA, hemophilia A; TGT, thrombin generation test; ABR: annual bleeding rate; VWF:Ag, antigenic Von Willebrand Factor levels; VWF:RCo, Von Willebrand Factor ristocetin cofactor activity; ETP, endogenous thrombin potential; NS, not significant.

**Table 5 jcm-11-03345-t005:** Clinical characteristics of the HA patients according to the group of *F8* mutations. (**A**) Identified mutations and ABR score based stratification of HA patients. (**B**) Clinical characteristics of HA patients as stratified based on the *F8* identified mutations and ABR score.

(A)						
Group		ABR PreTGT	Identified Mutation			
Group 1		≥1	p.Val115Ala (*n* = 2), p.Glu291Lys (*n* = 1), p.Ser308Leu (*n* = 1), p.Arg355 * (*n* = 1), p.Ser693Leu (*n* = 4), p.Phe1710Leu (*n* = 1), p.Lys1732Arg (*n* = 1), p.Gln2208Glu (*n* = 1), p.Thr2264dup (*n* = 1), deletion exons 2–7 (*n* = 1), deletion exon 6 (*n* = 1), deletion intron 6 and exon 9 (*n* = 1), deletion exons 12–13 (*n* = 2), deletion exon 26 (*n* = 1), intron 22 inversion (*n* = 4)			
Group 2		0	p.Thr74= (*n* = 1), p.Glu132Asp (*n* = 1), p.Ile192Thr (*n* = 1), p.Arg355 * (*n* = 1), p.Arg550Cys (*n* = 1), p.Ile567Thr (*n* = 1), p.Ser635Thr (*n* = 1), p.Gln645Serfs *8 (*n* = 1), p.Ser693Leu (*n* = 2), p.His1066Serfs *2 (*n* = 1), p.Trp1127 * (*n* = 1), p.Phe1710Leu (*n* = 1), p.Arg2016Gln (*n* = 1), p.Cys2040Gly (*n* = 1), p.Arg2178Cys (*n* = 2), p.Thr2264dup (*n* = 1), intron 22 inversion (*n* = 2), deletion exon 26 (*n* = 1)			
Group 3		≤1	No identified mutations (*n* = 10)			
**(B)**						
					** *p* ** **-Value ***	
**Characteristics Value**	**Group 1** **(*n* = 23)** **(A)**	**Group 2** **(*n* = 21)** **(B)**	**Group 3** **(*n* = 10)** **(C)**	**A vs. B**	**A vs. C**	**B vs. C**
Severity **						
1	6 (26.1)	13 (61.9)	10 (100.0)			
2	6 (26.1)	1 (4.8)	0 (0.0)			
3	11 (47.8)	7 (33.3)	0 (0.0)			
Age, years	36.0 (28.0–48.0)	31.0 (20.0–48.5)	45.0 (31.0–68.8)	NS	NS	NS
FVIII at TGT	10.0 (4.3–17.5)	16.7 (6.6–26.9)	28.6 (22.8–34.7)	NS	**<0.01**	NS
Paired FVIII diagnosis	2.5 (0.0–6.1)	10.0 (0.9–21.4)	26.0 (22.8–32.7)	NS	**<0.001**	**<0.05**
ABR preTGT	2 (1–3)	0 (0–0)	0 (0–0.3)	**<0.001**	**<0.001**	NS
ABR postTGT	1 (0–1.3)	0 (0–0)	0 (0–0.3)	**<0.05**	**<0.05**	NS
ETP (%)	62.0 (43.7–80.9)	72.8 (55.7–103.5)	91.5 (61.8–104.1)	NS	NS	NS
Peak (%)	37.6 (28.8–46.9)	45.1 (33.7–69.3)	58.7 (44.8–95.1)	NS	**<0.05**	NS
Time to peak (ratio)	1.6 (1.3–1.7)	1.6 (1.3–2.0)	1.5 (1.3–1.7)	NS	NS	NS
Lag time (ratio)	1.0 (0.7–1.3)	1.0 (0.8–1.3)	1.4 (1.1–1.5)	NS	**<0.05**	NS
Start tail (ratio)	1.5 (1.4–1.7)	1.4 (1.3–1.7)	1.3 (1.1–1.5)	NS	NS	NS

Continuous variables are presented as median and 1st–3rd quartile. FVIII:C and FVIII:Chr were measured at diagnosis and on the day of TGT. In patients under treatment, measurements were performed in the sample collected 48–72 h after FVIII administration. * Kruskal–Wallis non-parametric Test (Bonferroni correction for multiple comparisons), significant results are in bold. ** Severity: 1 Mild; 2 Moderate; 3 Severe. HA, hemophilia A; TGT, thrombin generation test; ABR: annual bleeding rate; ETP, endogenous thrombin potential; NS, not significant.

**Table 6 jcm-11-03345-t006:** Clinical characteristics of the HA patients according to the treatment received.

	Treatment (*n* > 3)				*p*-Value *					
Characteristics Value	Elocta^®^ (*n* = 20) (A)	Nuwiq^®^ (*n* = 9) (B)	Kovaltry^®^ (*n* = 4) (C)	DDAVP (*n* = 17) (D)	A vs. B	A vs. C	A vs. D	B vs. C	B vs. D	C vs. D
Severity **										
1	1 (5)	6 (66.7)	2 (50)	17 (100)						
2	5 (25)	0 (0.0)	2 (50)	0 (0)						
3	14 (70)	3 (33.3)	0 (0)	0 (0)						
Age, years	35.5 (30.3–47.8)	38.0 (10.0–41.5)	49.5 (35.3–70.5)	34.0 (12.5–48.5)	NS	NS	NS	NS	NS	NS
FVIII at TGT	8.9 (3.4–16.2)	15.8 (5.4–27.9)	4.6 (4.3–13.7)	28.2 (22.6–39.7)	NS	NS	**<0.001**	NS	NS	**<0.05**
FVIII at diagnosis (assay matched)	0.6 (0.0–2.8)	14.8 (1.5–24.1)	7.3 (3.9–15.0)	25.0 (11.0–35.2)	NS	NS	**<0.001**	NS	NS	NS
ABR preTGT	1 (0.0–2.0)	0.0 (0.0–1.5)	1.5 (0.3–2.0)	0.0 (0.0–0.0)	NS	NS	NS	NS	NS	NS
ABR postTGT	1 (0.0–1.8)	0 (0.0–1.0)	1 (0–2)	0.0 (0.0–0.0)	NS	NS	**<0.001**	NS	NS	NS
ETP (%)	60.5 (54.8–93.5)	97.7 (76.7–126.6)	67.8 (50.6–78.7)	67.0 (53.3–90.5)	NS	NS	NS	NS	NS	NS
Peak (%)	37.9 (29.0–55.3)	84.0 (46.0–95.2)	38.8 (32.0–45.0)	45.6 (31.2–62.3)	NS	NS	NS	NS	NS	NS
Time to peak (ratio)	1.7 (1.4–1.9)	1.5 (1.3–1.7)	1.5 (1.3–1.6)	1.6 (1.3–1.8)	NS	NS	NS	NS	NS	NS
Lag time (ratio)	1.1 (0.8–1.3)	1.2 (0.8–1.4)	0.9 (0.7–1.1)	1.1 (1.0–1.5)	NS	NS	NS	NS	NS	NS
Start tail (ratio)	1.5 (1.3–1.8)	1.4 (1.1–1.5)	1.6 (1.3–2.0)	1.4 (1.3–1.6)	NS	NS	NS	NS	NS	NS

Continuous variables are presented as median and 1st–3rd quartile. FVIII:C and FVIII:Chr were measured at diagnosis and on the day of TGT measurement. In patients under treatment, measurements were performed in the sample collected 48–72 h after FVIII administration. * Kruskal–Wallis non-parametric Test (Bonferroni correction for multiple comparisons), significant results are in bold. ** Severity: 1 Mild; 2 Moderate; 3 Severe. HA, hemophilia A; TGT, thrombin generation test; ABR: annual bleeding rate; ETP, endogenous thrombin potential; NS, not significant.

## Data Availability

All data is included in the Appendix A.

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
