# Peer review of "Applicability of the Thrombin Generation Test to Evaluate the Hemostatic Status of Hemophilia A Patients in Daily Clinical Practice"

_jcm, 2022, doi:10.3390/jcm11123345_

Round 1
Reviewer 1 Report
I thank the authors for the comprehensive improvement of the manuscript.
However I have still little comments.
Line 105 : You describe the results that will follow. I think it is not the right place.
Line 139 : Centrifugation at 4°C even for TGT is not correct. See the recommendation of Dargaud et al. in JTH 2017 DOI: 10.1111/jth.13743
REF 26 : use the last WFH recommendations 2021.
Author Response
We would like to thank Reviewer #1 for his/her constructive remarks. We answer below in a point-by-point fashion, and hope all the reviewer´s concerns are satisfactorily addressed. We think that the review process has continued to improve our manuscript, and hope it will be of interest to the readership of the Journal of Clinical Medicine.
Open Review 1
Comments and Suggestions for Authors
I thank the authors for the comprehensive improvement of the manuscript.
However I have still little comments.
Line 105 : You describe the results that will follow. I think it is not the right place.
We have removed this sentence from the Introduction, and have incorporated part of it in the Conclusions, as it was more or less repeated in the last sentence (see addition now in lines 509-510).
Line 139 : Centrifugation at 4°C even for TGT is not correct. See the recommendation of Dargaud et al. in JTH 2017 DOI: 10.1111/jth.13743
The reviewer is right. We had centrifuged samples at 4oC, and recently, we have modified the protocol so we maintain room temperature during the centrifugation steps. We mention this now in the Discussion section (see lines 484-487), and refer to the article mentioned by the reviewer in that remark. The reference was already included (now #44). We are certain that this has not affected the results, as both control and patient samples were handled in the same way.
REF 26 : use the last WFH recommendations 2021.
We have verified this, and we have referred to the last version already; it is the one dated in 2020 (now ref #35). See the website for confirmation:
https://guidelines.wfh.org/
“These materials are derived from the WFH Guidelines for the Management of Hemophilia, 3rd edition [© 2020 World Federation of Hemophilia. Haemophilia © 2020 John Wiley & Sons Ltd]”.

Reviewer 2 Report
The goal of the study was to validate the ST Genesia® automated thrombin generation analyzer to quantify relative haemostatic status in HA patients in a cohort of 54 patients. This work is substantial and well presented.
Although the approach is interesting (TGT is a global test that would better reflect the individual haemostatic status), authors should better explain why the equate severe hemophiliacs to moderate ones, or patients who bled with patients who did not. It would be interesting to specify it patients who bled had lower TGT compared to patients who did not, so that their treatment could be adjusted accordingly. By grouping everything, this data is not exploited. This would be one of the advantages of TGT over simple assays as it is supposed reflect the haemostatic status of the patient. This does not appear in the manuscript, and this point should be further discussed.
Moreover the categories Pré-TGT and Post TGT are not well defined and are not understandable to me
Author Response
We would like to thank Reviewer #2 for his/her constructive remarks. We answer below in a point-by-point fashion, and hope all the reviewer´s concerns are satisfactorily addressed. We think that the review process has continued to improve our manuscript, and hope it will be of interest to the readership of the Journal of Clinical Medicine.
Open Review 2
Comments and Suggestions for Authors
The goal of the study was to validate the ST Genesia® automated thrombin generation analyzer to quantify relative haemostatic status in HA patients in a cohort of 54 patients. This work is substantial and well presented.
Although the approach is interesting (TGT is a global test that would better reflect the individual haemostatic status), authors should better explain why the equate severe hemophiliacs to moderate ones, or patients who bled with patients who did not. It would be interesting to specify it patients who bled had lower TGT compared to patients who did not, so that their treatment could be adjusted accordingly. By grouping everything, this data is not exploited. This would be one of the advantages of TGT over simple assays as it is supposed reflect the haemostatic status of the patient. This does not appear in the manuscript, and this point should be further discussed.
As explained in the previous revision, we pooled Moderate and Severe patients because, after performing all the analyses with them separated, we realized that there was no variable in the study that was significantly different amongst Moderate and Severe groups. Considering the low N of Moderate, we decided to pool the groups to better value the clinical significance of TGT values (lines 192-195).
As shown in Table 4, the data already shows that the TGT parameters are different when comparing patients with ABR = 0 (non-bleeding) with patients with ABR > 0 (bleeding). The bleeding patients have lower ETP, Peak, and the Start Tail is increased, compared to non-bleeding patients. However, the differences were not significant in this cohort because it is treated to control their hemorrhagic manifestations. As we mention in the Discussion, and we propose, we are continuing our studies to obtain TGT parameters in patients at a bleeding phase or in patients with other coagulopathies. Interestingly, in Table 5, we show that when adding different stratification levels, such as mutation type and ABR (bleeding), TGT parameters, in this case Peak and Lag Time, significantly distinguish patient groups.
This study was observational, and TGT results did not influence the clinical management of patients (as mentioned in the text, line 359), but we propose in this manuscript that it will be a valuable tool in order to finely-tune treatment adjustment in a personalized manner in the future.
Moreover the categories Pré-TGT and Post TGT are not well defined and are not understandable to me
As explained in the previous revision, and in the manuscript (lines 120-123), “For each patient, the annual bleeding rate (ABR) (bleeds/year) was calculated as the number of bleeding events during 2 consecutive years divided by 2. ABR was calculated for 2017-2018 (preTGT), and 2018-2019 (postTGT)”.
The change in ABR pre- and post-TGT just reflects the cycle and wellness of the cohort. It is a quite descriptive variable in this case. Those patients that required a treatment adjustment, either due to bleeding or because they were newly diagnosed and consequently treated, improved globally the hemorrhagic profile of patients. However, there were significant differences only in the mild group (comparing pre- and post-TGT ABR, Table 2).

Reviewer 3 Report
Scandinavian and French studies have shown that at this moment the CAT technique is of an equal accuracy and reproducibility as most other haematological lab values, F.VIII levels included. The full automatisation of STGenesia remains however an important asset.
The ability of a TG test to discern haemophilia is completely dependent upon the concentration of tissue factor used. TF concentration in the STG-BLS reagent is unknown however. The results of this study therefore are dependent upon a manufacturing secret. This should be mentioned in the discussion.
It is not always immediately clear whether severe-moderate-mild refers to the factor VIII levels or to te clinical picture.
Aside: It remains slightly frustrating for the originator of modern TG techniques to see his contributions only indirectly referred to via an occasional review article authored by others.
Author Response
We would like to thank Reviewer #3 for his/her constructive remarks. We answer below in a point-by-point fashion, and hope all the reviewer´s concerns are satisfactorily addressed. We think that the review process has continued to improve our manuscript, and hope it will be of interest to the readership of the Journal of Clinical Medicine.
Open Review 3
Comments and Suggestions for Authors
Scandinavian and French studies have shown that at this moment the CAT technique is of an equal accuracy and reproducibility as most other haematological lab values, F.VIII levels included. The full automatisation of STGenesia remains however an important asset.
The ability of a TG test to discern haemophilia is completely dependent upon the concentration of tissue factor used. TF concentration in the STG-BLS reagent is unknown however. The results of this study therefore are dependent upon a manufacturing secret. This should be mentioned in the discussion.
We have mentioned this in the Discussion (lines 487-492). “Additionally, we should consider that the ability of a TGT to discern hemophilia is dependent on the concentration of TF used. As one of the provided reagents of the ST Genesia® platform, the concentration of TF remains a manufacturing secret, however, the amount of TF in the provided reagent has been optimized in order to minimize the potential procoagulant interference from contact phase activation during blood draw”.
It is not always immediately clear whether severe-moderate-mild refers to the factor VIII levels or to te clinical picture.
As we have mentioned in the Methods section, patients were classified at diagnosis, based on FVIII activity levels (lines 115-117). Accordingly, the clinical picture follows the diagnosis, and bleeding rate and required treatment regimens associate to respective diagnosis groups (more bleeding events in moderate and severe patients, and a higher proportion of “On demand” treatment in moderate and severe patients).
Aside: It remains slightly frustrating for the originator of modern TG techniques to see his contributions only indirectly referred to via an occasional review article authored by others.
We apologize to the reviewer. Actually, the reviewer is right, and it is not good practice to refer to Reviews solely. There are very useful Review works, because they are comprehensive, but we acknowledge this fault on our side. We have included original references in the Introduction section, line 76 (refs #8-9 and #11-15) and line 101 (ref #28).

This manuscript is a resubmission of an earlier submission. The following is a list of the peer review reports and author responses from that submission.
Round 1
Reviewer 1 Report
Global Haemostasis tests are a big issue in Haemophilia monitoring and follow up.
In this paper the authors show how TGA can help for profiling patients apart from FVIII concentration.
A control population group comparable for age could be advisable because hypercoagulable condition could characterized older people .
In my opinion a best clear message will be provided by splitting severe and moderate patients and patients on prophylaxis from on demand to assess correlation and prognostic value of TGT in predicting bleeding . A comparison between a stable FVIII concentration above 10% and a trough level of 4,6% can be not differentiated by TGT parameters , but are different in terms of bleedings. This could be a limit of TGT and not an advantage.
It could be useful to have some more informations on type of bleeding ( spontaneous , traumatic etc )
How can authors explain the change in ABR in pre e post TGT?
Could authors explain better how they have calculated ABR? Which are years taken in consideration? ( 2017-2018-2019 and one more?)
As regards mutations and bleedings, can authors explain how TGT can predict better than FVIII bleeding risk? Patients with same mutation have different ABR , but TGT cannot distinguish the different bleeding risk profile.
Reviewer 2 Report
|
This manuscript addresses the potential use of Thrombin Generation measured by Genesia in hemophilia. As mentioned by the authors this test is a promising tool to better assess the clinical bleeding profile than classical FVIII measurement and to tailor treatments.
However the study is a little bit confusing, in substance and in form.
First of all this paper needs major comprehensive revision before being considered for publication. In fact, besides the fact that English needs to be reviewed, the chart of tested patients is a little bit confusing. The cohort mixes severe, moderate, and mild patients, with patient on prophylactic or on demand in the same analyzed groups. It might be easier to analyze a more homogeneous group (only severe patient on prophylactic treatment, or only patient on “on-demand”). And to compare the results of FVIII, TGT, ABR and treatment in these more homogeneous cohort. Moreover the text, tables and figures are not comprehensive.
Here some reflections:
The abstract might contain less theoretical information and contain some elements of the results, including the “n” of the testes cohort. Line 45 : may is in bold ?
The introduction Line 84 : unit for ETP is not ok. Line 89 : you don’t mention velocity as a parameter of interest in TGT. Here and later, there are lot of information about Startail that is not of major importance. Maybe focus only on ETP and Peak ? Line 94 : why is CAT “not suitable to study several individual plasmas at a time” ??? For Genesia, you don’t mention the use of a reference plasma, a control of temperature and standardized reagents to improve reproducibility. At the end of the introduction, the aim of the study is not clear and is mixed with some information for Material and methods, and with conclusion ‘data. The aim should be clarified and the manuscript should focus on it. “identify a bleeding profile” ?
Materials and methods Study subjects : paragraph not clear. What is an oral consent ?? Why not sampling at the valley ? just before the following injection of the prophylactic scheme ?
Sample preparation : it is better to centrifuge at room temperature, not at 4°C.
The choice of FVIII clotting of chromogenic test might be explain.
Results
Demographic datas lack information about the dose ( dose/kg), the number of years under treatment. For ABR, what kind of bleeding was considered? Minor bleeding are included ? The Figure 2 is not relevant. Results are mixing patients under prophylactic and on-demand treatment, so the profile is very different. As already mentioned, a more homogeneous cohort could be of interest and easier to analyze. Tables and figures are not clear. Sometimes results include also discussion points. ABR post TGT is not clear : is there any intervention or only on observation ?
Discussion There are limitations due to the design of the study. (severe and moderate, prophylactic with on-demand, time of sampling). |